# A comprehensive approach to predicting weight gain and therapy response in psychopharmacologically treated major depressed patients: A cohort study protocol

**Maria S. Simon**◉*, **Barbara B. Barton, Catherine Glocker, Richard Musil**◉

Department of Psychiatry and Psychotherapy, University Hospital, Ludwig-Maximilians-University, Munich, Germany

* Maria.Simon@med.uni-muenchen.de

## Abstract

### Background

A subgroup of patients with Major Depressive Disorder shows signs of low-grade inflammation and metabolic abberances, while antidepressants can induce weight gain and subsequent metabolic disorders, and lacking antidepressant response is associated with inflammation.

### Objectives

A comprehensive investigation of patient phenotypes and their predictive capability for weight gain and treatment response after psychotropic treatment will be performed. The following factors will be analyzed: inflammatory and metabolic markers, gut microbiome composition, lifestyle indicators (eating behavior, physical activity, chronotype, patient characteristics (childhood adversity among others), and polygenic risk scores.

### Methods

Psychiatric inpatients with at least moderate Major Depressive Disorder will be enrolled in a prospective, observational, naturalistic, monocentric study using stratified sampling. Ethical approval was obtained. Primary outcomes at 4 weeks will be percent weight change and symptom score change on the Montgomery Asberg Depression Rating Scale. Both outcomes will also be binarized into clinically relevant outcomes at 5% weight gain and 50% symptom score reduction. Predictors for weight gain and treatment response will be tested using multiple hierachical regression for continuous outcomes, and multiple binary logistic regression for binarized outcomes. Psychotropic premedication, current medication, eating behavior, baseline BMI, age, and sex will be included as covariates. Further, a comprehensive analysis will be carried out using machine learning. Polygenic risk scores will be added in a second step to estimate the additional variance explained by genetic markers. Sample size calculation yielded a total amount of N = 171 subjects.

**Data Availability Statement:** No datasets were generated or analysed in the current protocol. A minimal anonymized data set will be uploaded to a

publicly accessible repository upon publication of data evaluations produced by this project.

**Funding:** The authors received no specific funding for this work.

**Competing interests:** I have read the journal's policy and the authors of this manuscript have the following competing interests: RM has received reimbursement by Janssen-Cilag, Emalex Biosciences, Boehringer-Ingelheim, and Oryzon for carrying out studies, and has received speaker's honoraria from Otsuka over the past five years, all outside the submitted work. No other potential or actual financial or non-financial competing interests apply. MSS, BB, and CG have declared that no competing interests exist. This does not alter our adherence to PLOS ONE policies on sharing data and materials.

## Discussion

Patient and physician expectancies regarding the primary outcomes and non-random sampling may affect internal validity and external validity, respectively. Through the prospective and naturalistic design, results will gain relevance to clinical practice. Examining the predictive value of patient profiles for weight gain and treatment response during pharmacotherapy will allow for targeted adjustments before and concomitantly to the start of treatment.

## Introduction

The complex pathophysiology of Major Depressive Disorder (MDD) and biological actions of antidepressant treatment regimen are still only partially understood. Newer trends in research depict this disorder as an inflammatory and metabolic disease [1, 2]. At least a substantial subgroup of patients shows chronic low-grade inflammation and metabolic abberances like overweight/obesity, metabolic syndrome, or cardiovascular disturbances [3, 4]. So far, it is well-known that antidepressant drug regimen can induce weight gain and further metabolic disorders [5, 6]. However, again only a subgroup of patients experiences such weight gain [5]. Further, antidepressant response is lacking in about one-third of patients and has been associated with pro-inflammatory activation [7, 8]. As there is a substantial overlap between inflammation, cardiometabolic disorders, and depression, it is interesting to raise attention to contributing predictive factors.

In more detail, markers of atherosclerosis, cardiovascular disease (CVD), and inflammation include circulating compounds such as C-reactive protein (CRP), fibrinogen, immunoglobulins, adhesion molecules, and pro-inflammatory cytokines (e.g., macrophage migration inhibitory factor (MIF)), and most of them are dysregulated in depressed patients [9, 10]. Noteworthy, many studies investigating inflammation in psychiatric disease exclude the presence of any diagnosed immunologic or at least uncontrolled cardiovascular diseases. Thus, low-grade abnormalities are already present even without the presence of manifest diseases. Prospectively, depressed patients have a high risk for developing CVD during lifetime [11]. A positive relationship between body mass index (BMI) and CRP levels in MDD indicated that patients with higher BMI already show some inflammatory activation [4]. In obesity, adipose tissue promotes low-grade inflammation [12], for which it seems an important component for understanding the relationship between body weight, inflammation, and metabolic alterations. It is now interesting to disentangle whether overweight as a precursor of metabolic disturbances is present in a group of patients per se and whether weight-gain associated antidepressant drug regimen may be an accelerator of such a development. Another emerging component in the metabolic-inflammatory machinery is the gut microbiome for which first indications of an imbalance in depression exist: the gut-brain-axis links the enteric with the central nervous system, whereas gut microbiome has been shown to play a crucial role also in inflammatory processes and depression [13, 14].

During antidepressant therapy, patients may experience a weight gain of 0.83kg up to 2.73kg within the first 12 weeks depending on the drug [15]. Typical weight-gain associated antidepressants are paroxetine, mirtazapine, doxepin, amitriptyline, and citalopram [15–17]. Alongside the treatment with antidepressants, MDD therapy strategies include augmentation with antipsychotics and mood stabilizers. A study investigating weight gain after antidepressant, mood stabilizer, and antipsychotic intake revealed that a weight gain of at least 5% within the first four weeks of treatment was predictive for further substantial weight gain after 12

months [18]. Thus, early weight gain already serves as a reliable long-term indicator and should be addressed [18]. From antipsychotic treatment, evidence shows that olanzapine has been associated with an increase of pro-inflammatory cytokines, and this low-grade inflammation was in turn associated with an increase of adipose tissue [19, 20]. Since it is known that inflammatory activation is associated with treatment resistance [21], though this seems drug class dependent [22], some interest has also been paid to metabolic disturbances and treatment response in depressed patients. For instance, presence of metabolic syndrome, low levels of HDL cholesterol, hypertriglyceridemia, hyperglycemia, and higher levels of interleukin-6 were found to be associated with depression chronicity in antidepressant users [23, 24]. Thus, it would be interesting to investigate whether low-grade inflammatory (and already present metabolic) abnormalities serve as predictors for weight gain during pharmacotherapy and treatment response. Further, first results indicate a relationship between microbiota composition and response or remission to antidepressants [25, 26]. As for the side effect of weight gain, antipsychotic-induced weight gain was shown to be modulated by anti- and prebiotic use [27–30]. Thus, microbiome composition may also contribute to predicting response and weight gain in antidepressant users. Other factors that have been associated with weight gain in previous studies are eating behavior, circadian rhythm, and genetic risk: Previous analyses show that eating behavior before the start of treatment was predictive for weight gain [31, 32], while the increase of dysfunctional eating behavior (disinhibition of eating/ emotional eating) during treatment phase also predicted an increase of body weight after treatment, independently and in interaction with drug dosage [33, unpublished observations]. Further, it is being proposed that the circadian system is involved in the development of drug-induced metabolic disturbances [34] and that disturbances of the circadian system and irregular food intake (against the individual circadian rhythm) pose risk factors for weight gain and metabolic disorders even without taking medication into account [35–37]. In a study with schizophrenic patients, polygenic risk scores of schizophrenia and obesity were found to be predictive of antipsychotic- induced weight gain [Franz et al., 2021, personal communication June 15, 2021]. This may probably also apply for depressed patients on antidepressant drug regimen.

Taken together, some studies point to the involvement of immuno-metabolic processes in the development of metabolic side effects and for antidepressant responsiveness. However, so far only a fragmented picture exists based on scarce evidence. Therefore, it would be interesting to study comprehensively whether there are signs and early-stage abnormalities regarding adipokine and cytokine activity, microbiome composition, eating behaviors, circadian rhythm, and polygenic risk in patients within the normal BMI range that already show predictive capability for weight gain during antidepressant therapy and treatment response.

## Objectives and hypothesis

Various inflammatory and metabolic components have been or are beginning to be studied in depressed patients to identify biological underpinnings of the disease, to understand treatment side effects, and subsequently to improve tailored treatment regimens. Evidence from literature on associations between diverse biological markers point to their co-existance in a certain patien group (see introduction). Here we aim at a comprehensive phenotyping of this patient group while studying the relation of such phenotypes to weight gain as a side effect of pharmacotherapy and treatment response. We will therefore investigate the predictive capability of baseline plasma pro-inflammatory/-atherosclerotic biomarkers, blood lipid profile, microbiome composition, polygenic risk, body fat distribution, eating behavior and food intake, chronotype (as a proxy for individual circadian rhythm), sedentary behavior, atypical or melancholic type of depression, and history of childhood trauma regarding weight gain and

antidepressant response in a depressed patient sample. We will then use relevant predictors to create an algorithm for determining individual risk of early weight gain and treatment resistance. Furthermore, we will explore the relationships between all these parameters at baseline to deduce phenotypic patient profiles.

To develop the risk factor profile in MDD patients for adverse metabolic and clinical outcome, the following hypotheses will guide the analyses: a) A baseline profile consisting of elevated pro-inflammatory and aberrant metabolic signature, higher body fat, and higher proportion of gram-negative gut bacteria (unbalanced microbiome composition) characterizes a group of depressed patients. b) Patients without abnormal metabolic profile but pro-inflammatory signature, dysfunctional lifestyle (eating behavior, sedentary behavior), and a history of childhood trauma show a stronger body weight gain. c) Patients with a pro-inflammatory signature at baseline show no significant change of depressive symptoms after four weeks.

## Materials and methods

### Study design and setting

A prospective observational study will be conducted at the Department of Psychiatry and Psychotherapy, University Hospital Munich. The estimated duration of data acquisition is two years, prospectively starting from October 2021. Thus, the study is in the process of being initialized and recruitment has not yet started. Since we are aiming at deducing recommendations for daily clinical routine, a naturalistic approach including a broad patient collective and without specification of particular drugs was chosen. However, some restriction criteria need to be applied to achieve reasonable internal validity. To achieve balance of factors that likely bias results, we will stratify sampling for antidepressant treatment prior to/at hospitalization and sex. This will also allow reasonable sample size for potential comparisons or subgroup analyses.

### Study procedures

The study protocol was approved by the ethical committee of the medical faculty, Ludwig-Maximilians-University Munich (project-nr. 21–0357; approval date June 15, 2021). To ensure safety of patients and patient-related data, research will comply with international and national conventions and declarations, i.e. the standards of good clinical practice, the Declaration of Helsinki (and its latest revisions), and regulations regarding data protection (DSGVO). The main assessments will be done at baseline and at four weeks of observation period under antidepressant treatment. This period will usually start at hospital admission and at the latest within two weeks after admission immediately after the onset of medication. Many physiological measures will be collected via our in-house large-scale biobanking system. Else data will be collected during personal visits at baseline, week four, and further for two weeks after discharge per post. Biobanking and additional study-related measurements will be timely coordinated. In the case of a delay of more than 5 days between biobanking (metabolic determinations) and other physiological determinations, the first will be repeated. Fig 1 gives an overview of the planned study flow.

### Assessments

Diagnosis of depression will be derived from the electronic health recording system of the clinic as entered by the treating physician at hospital admission. Further, diagnosis will be verified by the Structured Clinical Interview for DSM-IV Axis I Disorders (SCID-I, German version SKID-I) [38] by a trained rater. Demographic data (e.g., age, sex, prior psychotropic

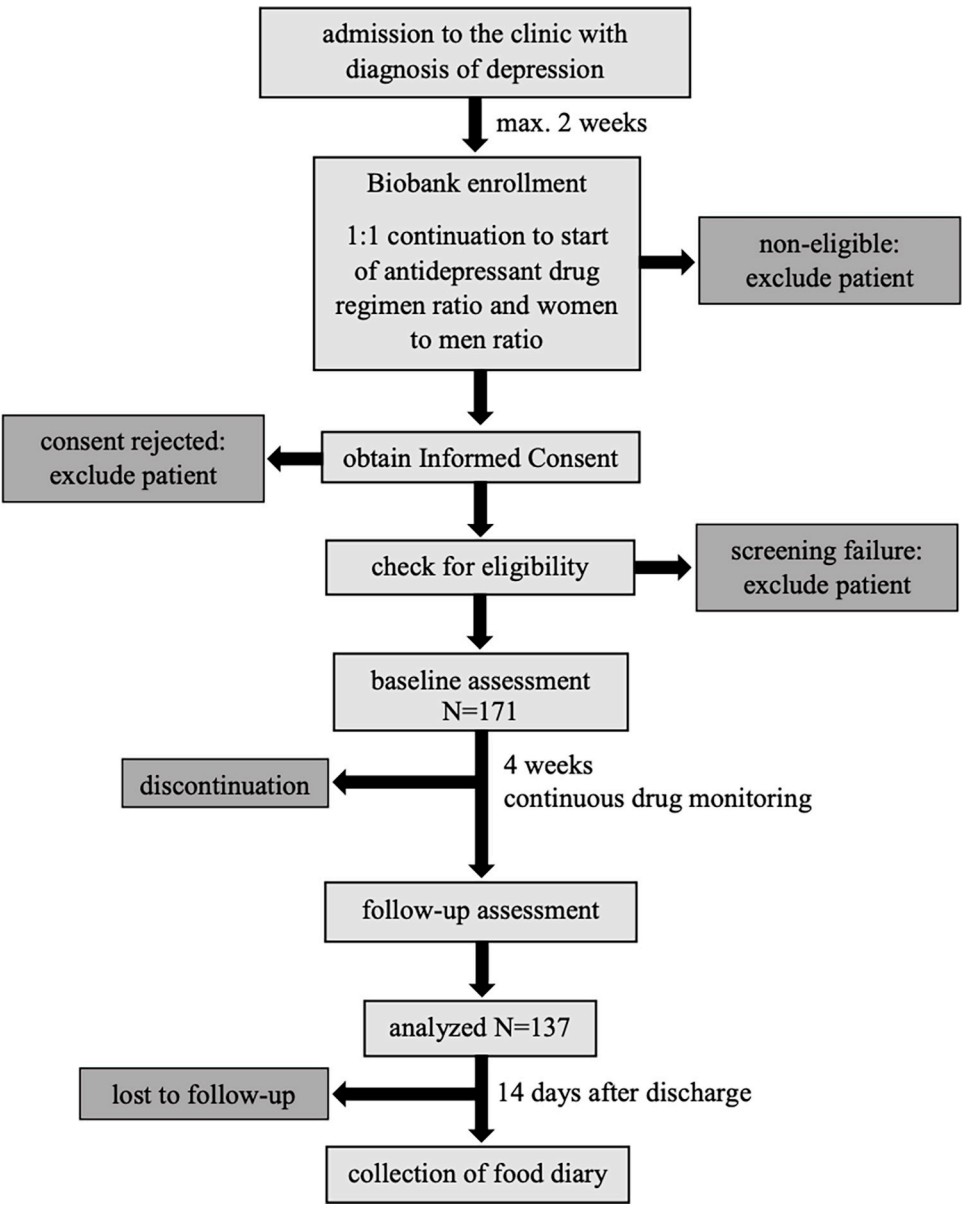

**Fig 1. Study flow.**

medication) will be inquired during standard patient admission procedure to the clinic. Body weight will be measured on a calibrated scale, BMI calculated, and waist circumference will be taken as standardized by the World Health Organization [39]. Further, continuous drug monitoring of antidepressant drug regimen and use of statins, non-steroidal anti-inflammatory drug (nonselective and selective cox-2-inhibitors), paracetamol and minocycline, as well as use of insulin/glucose-regulating and cardiovascular medications will be performed throughout the study phase using the electronic health recording system of the clinic. From blood samples, metabolic parameters will be determined: fasting total cholesterol, HDL-cholesterol, LDL-cholesterol, triglycerides, fasting blood glucose, HbA1c. Presence of metabolic syndrome and cardiovascular risk will be identified according to the International Diabetes Federation [40] and the PROCAM score [41], respectively. Further, circulating pro-inflammatory compounds and

adipokines will be determined from plasma by enzyme-linked immunosobent assay: CRP (R&D Systems® DCRP00), highsensitive interleukin-6 (R&D Systems® HS600C), highsensitive interleukin-1beta (R&D Systems® HSLB00D), highsensitive tumor necrosis factor alpha (R&D Systems® HSTA00E), soluable intracellular adhesion molecule 1 (Hölzl ELH-ICAM1-1), MIF (R&D Systems® DMF00B), leptin (R&D Systems® DLP00), ghrelin (abcam ab263887), osteopontin (R&D Systems® DOST00). Microbiome composition will be determined from stool samples by high throughput sequencing of bacterial 16S-rRNA genes using standardized method (DIN EN ISO 15189). A body composition analyzer will be used to assess body fat, amongst others. For the investigation of genetic markers, DNA will be extracted from whole blood and a genome-wide association study will be carried out to analyze single nucleotide polymorphisms, and polygenic risk score will be calculated. Genotyping will be done using Infinium Global Screening Array-24 Kit (Illumina, Inc).

The Inventory of Depressive Symptomatology self-rating version (IDS-SR) assesses DSM-IV diagnostic criteria of major depressive episode with 30 items and allows for categorizing melancholic and atypical depression [7, 42]. Drieling and colleagues [43] verified high internal consistency for a translated German version. Montgomery Åsberg Depression Rating Scale (MADRS) total score indicates depression severity based on 10 items and will be assessed by a trained interviewer [44, 45]. The translated German version showed high internal consistency [45]. MADRS was chosen as a primary outcome measure because it showed superior reliability over HAMD17 [46, 47], and response/remission criteria were confirmed [48]. Treatment response is defined by at least 50% symptom score reduction of the initial MADRS score, remission is defined by MADRS score of 7 or less at week 4. Percentaged score reduction in relation to baseline sum score will also be calculated. Presence of childhood adverse experience will be assessed using the 28-item self-rating Childhood Trauma Questionnaire [49] in the validated German version [50]. The German version of the three-factor eating questionnaire short form with 18 items (TFEQ, [51]; *Fragebogen zum Essverhalten* (FEV)), validated by Pudel and Westenhoefer [52]; adapted after Cappelleri and colleagues [53]) assess eating behavior. From the FEV three subscale sumscores can be derived: cognitive control, disinhibition (emotional eating), and hunger. To assess the sleeping behavior and chronotype the Munich Chronotype Questionnaire (MCTQ; as used in Roenneberg et al. [54, 55]) will be used. Further, the Regensburg Microbiome Questionnaire will be used (ReMBiQ) [T. Baghai, personal communication, September 4, 2020] to assess potential confounders on microbiome composition. In addition, the times of meal are recorded in a diary over the study period and within 14 days after discharge from the clinic in order to quantify the regularity of the subjects' food intake when they 1) have the choice of adhering or not to the clinical schedule and 2) are able to freely determine their own daily schedule at home. This will be done by recording the times of all meals, snacks, and caloric drinks in a simple table diary. Based on the scattering of meal and snack times, a score for the regularity of mealtimes can be established [37, 56, 57]. Physical activity will be measured by the SIMPAQ [58] which was validated for the mentally ill [59].

## Eligibility criteria

A selected group of hospitalized psychiatric patients diagnosed with MDD (depressive episode ICD-10 F32.x, recurrent depression F33.x) will be investigated. Patients may be hospitalized due to various reasons, e.g., installment or change of psychotropic medication, lack of self-reliant conduct of life, stressful home environment preventing recovery, etc. The following inclusion criteria apply:

1. male and female psychiatric inpatients aged between 18 and 60 years

2. ability to sufficiently speak and comprehend the German language

3. continuing or beginning antidepressant drug regimen within two weeks after admission

4. MADRS score $\geq$ 14

5. capability of providing written informed consent or in agreement with a legal representative.

   Subjects were excluded from participation if they met the following exclusion criteria:

1. cognitive disability impeding completion of self-rating questionnaires or answering questions reasonably;

2. presence of psychotic features, manic episode, bipolar disorder, disorders from the schizophreniform spectrum, eating disorder, drug or alcohol addiction. Other disorders (e.g., adjustment disorder, obsessive compulsive disorder, anxiety disorder, somatoform disorder, personality disorder) in case the symptoms predominate the clinical picture;

3. presence of uncontrolled systemic disease, uncontrolled somatic (other than metabolic or cardiovascular disturbances) / neurological disorder, current or recent (last month) severe infections or inflammatory disease, current or recent (last month) physical trauma;

4. treatment with electroconvulsive therapy;

5. patients with immediate risk for suicidal behavior;

6. pregnant, breast-feeding, or postmenopausal women.

## Recruitment and adherence

Patients will be recruited at the wards of the Department of Psychiatry and Psychotherapy, University Hospital Munich. All subjects will be asked to provide written informed consent prior to enrollment. Eligible patients will be enrolled as soon as diagnosis is secured, either directly after admission in case they are continuing prescribed antidepressant medication, or within the first two weeks after admission in case antidepressant treatment is established. To reduce missing data, the study personnel will encourage patients to adhere to the protocol, check completeness of data promptly after the scheduled visit, and collect missing data immediately.

## Outcomes

As primary outcomes, weight change in relation to baseline body weight, and depression symptom change at 4 weeks will be evaluated. Also, patients will be classified as weight-gainers and non-weight-gainers based on at least 5% weight gain in relation to body weight at baseline and as responders and non-responders. Biomarkers and patient characteristics will be evaluated as predictors. A further outcome are associative measures between pro-inflammatory and metabolic markers, body fat, and microbiome composition. Secondary outcomes will be the change of metabolic, inflammatory and microbiotic, and body composition profiles from baseline to week 4, as well as the associations among the biological variables with all other patient characteristics.

## Safety considerations

Since the natural treatment upon physician discretion is observed, no study-specific risks arise from recorded medication intake. Negative expectation towards regular treatment due to

engagement in topics like treatment resistance and slight exhaustion due to filling out questionnaires may contribute to potential non-adherence to the prescribed medication and study assessments. Further, hematoma due to blood withdrawal may occur.

## Data management

Part of the data will be retrieved from the biobanking system electronically and paper based. All data, including additional project-related assessments, will be stored centrally using the clinic´s standardized Biobank Management-System (BBMS) Software (CentraXX, Fa. Kairos, Bochum). Biomaterials will be stored in a -80˚C freezer until analysis. Data collected by paper forms (i.e. biobank form, rating scales) will be stored in patient ID-labeled binders in a lockable and only for study staff accessible place. Raw data will be pseudonymized and doubly pseudonymized when uploaded in CentraXX. A coding list of patient identity will be stored at the clinic study center. Data will be anonymized upon patient request or withdrawal of consent during the study. All data in paper form or electronic databases for evaluation purposes will be stored for 15 years after the end of the project at the clinic archive and password protected clinic drive, respectively. Digital data storage within CentraXX is not limited in time to enable further analyses for prospective research. A minimal anonymized data set will be uploaded to a publicly accessible repository upon publication of data evaluations produced by this project.

## Sample size and statistical analyses

Statistical analyses will mainly be performed using R. Descriptive statistics will be calculated for standard demographic data and clinical variables using frequencies for categorical and mean (standard deviation) for continuous data. In case of non-normality, median (interquartile range) will be reported. Primary analyses will be carried out at 4 weeks treatment duration. Predictors for weight gain and treatment response, baseline cytokines and adipokines of highest interest, will be tested using multiple hierachical regression for percent weight change and symptom score reduction, as well as multiple binary logistic regression for dichotomized weight gain and treatment response variables. Psychotropic pretreatment, medication during treatment, eating behavior, baseline BMI, age, and sex will be included as covariates. Other parameters like microbiome composition, body fat distribution, and patient characteristics will be analyzed similarly. To control for the amount of exposure to psychotropic medication intake throughout the study, we will develop a scoring system quantifying the medication-associated risk for weight gain which will be included as a variable in our analyses. Further, treatment exposure will be dichotomized to evaluate the Relative Risk for weight gain. Correlational analyses (Pearson or Spearman-Rho) or tests for difference (t-test or Mann-Whitney test) will be used to identify the associations between metabolic, inflammatory biomarkers, and body composition, and microbiota composition, as well as other patient characteristics. The course of parameters from baseline to week 4 will be investigated using linear regression of burden of medication intake during treatment on parameter change score including covariate adjustment for psychotropic pretreatment. Last, inflammatory biomarkers will be added to a support-vector machine learning model using neurominer [60], among previously explored patient variables like eating behavior, food craving, age, BMI, sex, and abdominal girth [31]. In a second step, the polygenic risk score retrieved from GWAS using the software plink version 1.9 will also be added to the model to estimate the additional variance explained by genetic markers. Complete case analysis will be carried out in case of less than 5% missing data. Otherwise, missing data will be imputed using multiple imputation method. Sample size calculation was based on estimated medium effect size (f2 = 0.15) for increase of explained variance of one additional predictor in the multiple hierarchical regression model with baseline predictors.

Power was set at 95% and alpha-level was set at 0.005 taking Bonferroni multiple test correction into account (nine parameters of highest interest tested in single models). This led to a sample size of N = 137 plus 25% (N = 34) to account for potential drop-outs. The total sample size consists of N = 171 subjects.

## Discussion

The study presented here will provide a comprehensive phenotyping of inherent and behavioral patient characteristics, as well as biological markers, in psychiatric inpatients with depression. This will enable us to carve out patient profiles and determine diagnostic subgroups. Furthermore, the study will provide evidence for the predictive value of such patient profiles for treatment response, thereby offering a basis for deducing recommendations for patient stratification regarding treatment choice. Last but not least, the study will provide an individually applicable prediction model for weight gain during pharmacotherapy, thus allowing for early counterregulating interventions. Noteworthy, the results may also shed more light on the question of whether weight gain is a sign of response, which is debated in the scientific community. Hypothetically, as overweight/obesity seem to be related to inflammation and treatment resistance, and individuals with low/normal BMI before medication experience weight gain, the normal-weight non-inflammatory individuals may represent the treatment- responsive subgroup.

Some limitations lie in the study design and are discussed in the following. First of all, due to a non-random sampling technique and eligibility criteria excluding psychiatric comorbidities for the most part, possible sampling bias has to be considered. Thus, results from the study sample may lack external validity with respect to the general hospital population. Therefore, results will be generalized with caution and the limitation by frequent major comorbidities will be discussed. Furthermore, acutely suicidal patients, who may represent a very severe subgroup of patients, are excluded. This may imply a limitation regarding the primary outcomes if disease severity was associated with weight gain or treatment response. Trends of a gradual relationship between these variables may be explored in the available sample. Other patient characteristics may also be imbalanced. Thus, a detailed descriptive analysis of the study sample will be provided. Further, the advantage of a medium to large sample size over a small one are increased chances for obtaining a representative sample. Another challenging aspect is the completeness of data. Systematic drop-out can bias the results, e.g., leaving the hospital before the follow-up assessment due to early treatment response will underestimate treatment success. To prevent biased data, we have chosen a short observation time for which the vast majority of patients stays at the clinic. Additionally, reasons for drop-out will be recorded. To avoid missings throughout the assessments, the study team will be trained to engage in close monitoring of data completeness and in motivating patients. Since the study is of observational nature, no blinding will be implemented which may lead to subject and physician expectations regarding the primary outcomes. In practice, treatment follows the discretion of physicians, who are committed to provide the best therapy possible.

Our study design also incorporates several strengths. First, the prospective approach allows for an estimation of risk for outcome development and therefore of the impact in clinical practice.

Second, stratification for the two important confounding variables sex and pretreatment enables balanced analyses. Third, a continuous score for treatment exposure prevents unbalanced treatment groups and permits high power statistical analyses (linear regression). Furthermore, this score accounts for potential bias due to non-adherence to prescribed treatment, especially when facing unwanted side effects or lacking response, as it measures the actual

medication taken. This also increases generalizability to clinical practice, where patients are likely to discontinue or switch to other treatments.

The results provide new insights into the biological underpinnings of depression and their interplay with psychotropic therapy, enabling clinicians to identify patients at high risk before start of treatment, and offering guidance for potential strategies to improve side effect management and treatment success early on.

## Acknowledgments

We would like to acknowledge the contribution of the following people who supported the development of the protocol: Dr. Dominic Landgraf, Prof. Dr. Nikolaos Koutsouleris, Dr. Sergi Papiol, Prof. Dr. Andrea Schmitt, and Prof. Dr. Peter Falkai. Language editing was assisted by Thelma Coutts.

## Author Contributions

**Conceptualization:** Maria S. Simon, Barbara B. Barton, Catherine Glocker, Richard Musil.

**Methodology:** Maria S. Simon, Barbara B. Barton, Catherine Glocker, Richard Musil.

**Project administration:** Maria S. Simon, Richard Musil.

**Writing – original draft:** Maria S. Simon.

**Writing – review & editing:** Barbara B. Barton, Catherine Glocker, Richard Musil.

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
