## [Decision Letter · Decision Letter 0]

29 Apr 2022

PONE-D-21-24585A comprehensive approach to predicting weight gain and therapy response in psychopharmacologically treated major depressed patients: a cohort study protocol.PLOS ONE

Dear Dr. Simon,

Thank you for submitting your manuscript to PLOS ONE. After careful consideration, we feel that it has merit but does not fully meet PLOS ONE’s publication criteria as it currently stands. Therefore, we invite you to submit a revised version of the manuscript that addresses the points raised during the review process.

Please see the comments from one reviewer below. Please note that we have only been able to secure a single reviewer to assess your manuscript. We are issuing a decision on your manuscript at this point to prevent further delays in the evaluation of your manuscript. Please be aware that the editor who handles your revised manuscript might find it necessary to invite additional reviewers to assess this work once the revised manuscript is submitted. However, we will aim to proceed on the basis of this single review if possible. 

We look forward to receiving your revised manuscript.

Kind regards,

Hanna Landenmark

Staff Editor

PLOS ONE

Journal Requirements:

3. Thank you for stating the following in the Competing Interests section: "I have read the journal's policy and the authors of this manuscript have the following competing interests: RM has carried out studies financed by Janssen-Cilag, Emalex, Boehringer-Ingelheim and has received speaker’s honoraria from Otsuka over the past three years. MSS, BB, and CG have declared that no competing interests exist."

We note that you received funding from a commercial sources: Janssen-Cilag, Emalex, Boehringer-Ingelheim and Otsuka.

Reviewers' comments:

Reviewer's Responses to Questions

**Comments to the Author**

1. Does the manuscript provide a valid rationale for the proposed study, with clearly identified and justified research questions?

Reviewer #1: Yes

2. Is the protocol technically sound and planned in a manner that will lead to a meaningful outcome and allow testing the stated hypotheses?

Reviewer #1: Yes

3. Is the methodology feasible and described in sufficient detail to allow the work to be replicable?

Reviewer #1: Yes

4. Have the authors described where all data underlying the findings will be made available when the study is complete?

Reviewer #1: Yes

5. Is the manuscript presented in an intelligible fashion and written in standard English?

Reviewer #1: Yes

6. Review Comments to the Author

You may also provide optional suggestions and comments to authors that they might find helpful in planning their study.

Reviewer #1: Introduction, Objectives and hypotheses

The authors indicate that they will evaluate circadian rhythms, but it seems more pertinent to indicate that they will only evaluate the chronotype (Lines 136 and 151).

Objectives and hypothesis

They indicate three main hypotheses, in the first of these (lines 157-159), they indicate “a depressed patient subgroup. They mean that they will evaluate the subtypes of depression? In any case, to which subgroup do they refer?

I do not find in the introduction the support of this hypothesis.

Materials and Methods

Eligibility criteria

It would be useful if they indicated what are the criteria for hospitalization "psychiatric inpatients"

Not all readers are familiar with psychiatric treatments, define acronyms for ECT.

Outcomes

The authors only indicate two outcomes: “body weight” and “depression symptom change”, but it is not clear what response they will evaluate with respect to the first hypothesis (lines 157-159) when referring to “a depressed patient subgroup”.

7. PLOS authors have the option to publish the peer review history of their article (what does this mean?). If published, this will include your full peer review and any attached files.

Reviewer #1: No

---

## [Author Response · Author response to Decision Letter 0]

12 Jun 2022

Editorial comments:

Style was checked again and is in compliance with the requirements.

As is appears from the author guidelines, the funding information and the financial disclosures mean the same section. The authors received no specific funding for this work. Initially, we gave information about a planned funding application. However, no funding was installed so far by any third party. Thus, there are no grant numbers to report. So, we just leave it to the given statement. 

3. Thank you for stating the following in the Competing Interests section: "I have read the journal's policy and the authors of this manuscript have the following competing interests: RM has carried out studies financed by Janssen-Cilag, Emalex, Boehringer-Ingelheim and has received speaker’s honoraria from Otsuka over the past three years. MSS, BB, and CG have declared that no competing interests exist." We note that you received funding from a commercial sources: Janssen-Cilag, Emalex, Boehringer-Ingelheim and Otsuka.

Please provide an amended Competing Interests Statement that explicitly states this commercial funder, along with any other relevant declarations relating to employment, consultancy, patents, products in development, marketed products, etc. Within this Competing Interests Statement, please confirm that this does not alter your adherence to all PLOS ONE policies on sharing data and materials by including the following statement: "This does not alter our adherence to PLOS ONE policies on sharing data and materials.”

If there are restrictions on sharing of data and/or materials, please state these. Please note that we cannot proceed with consideration of your article until this information has been declared. Please include your amended Competing Interests Statement within your cover letter. We will change the online submission form on your behalf.

We have now amended the competing interests statement as follows:

I have read the journal's policy and the authors of this manuscript have the following competing interests: RM has received reimbursement by Janssen-Cilag, Emalex Biosciences, Boehringer-Ingelheim, and Oryzon for carrying out studies, and has received speaker’s honoraria from Otsuka over the past five years, all outside the submitted work. No other potential or aactual financial or non-financial competing interests apply. MSS, BB, and CG have declared that no competing interests exist. This does not alter our adherence to PLOS ONE policies on sharing data and materials.

4. In your Data Availability statement, you have not specified where the minimal data set underlying the results described in your manuscript can be found. PLOS defines a study's minimal data set as the underlying data used to reach the conclusions drawn in the manuscript and any additional data required to replicate the reported study findings in their entirety. All PLOS journals require that the minimal data set be made fully available. For more information about our data policy, please see http://journals.plos.org/plosone/s/data-availability. Upon re-submitting your revised manuscript, please upload your study’s minimal underlying data set as either Supporting Information files or to a stable, public repository and include the relevant URLs, DOIs, or accession numbers within your revised cover letter. For a list of acceptable repositories, please see http://journals.plos.org/plosone/s/data-availability#loc-recommended-repositories. Any potentially identifying patient information must be fully anonymized. Important: If there are ethical or legal restrictions to sharing your data publicly, please explain these restrictions in detail. Please see our guidelines for more information on what we consider unacceptable restrictions to publicly sharing data: http://journals.plos.org/plosone/s/data-availability#loc-unacceptable-data-access-restrictions

Note that it is not acceptable for the authors to be the sole named individuals responsible for ensuring data access. We will update your Data Availability statement to reflect the information you provide in your cover letter.

Since the present manuscript is a study protocol, no data was generated and evaluated in this manuscript. Thus, there is no data that can be made available. However, the authors state that upon later publication of data evaluations from this project, the minimal data set will be uploaded to a publicly accessible repository. The statement was changed in the text.

5. We note that you have indicated that data from this study are available upon request. PLOS only allows data to be available upon request if there are legal or ethical restrictions on sharing data publicly. For more information on unacceptable data access restrictions, please see http://journals.plos.org/plosone/s/data-availability#loc-unacceptable-data-access-restrictions

b) If there are no restrictions, please upload the minimal anonymized data set necessary to replicate your study findings as either Supporting Information files or to a stable, public repository and provide us with the relevant URLs, DOIs, or accession numbers. For a list of acceptable repositories, please see http://journals.plos.org/plosone/s/data-availability#loc-recommended-repositories. We will update your Data Availability statement on your behalf to reflect the information you provide.

Please see the answer for comment number 4.

There are no Supporting Information files with this submission.

The reference list was checked again. One reference was updated as it was accepted for publication in the meantime (reference number 33).

Reviewer comments: 

Thank you very much for your review and pointing out unclarities. We have now revised the manuscript according to the comments by improving explanations in the appointed sections.

1. Introduction, Objectives and hypotheses: The authors indicate that they will evaluate circadian rhythms, but it seems more pertinent to indicate that they will only evaluate the chronotype (Lines 136 and 151). 

Indeed, in the introduction we speak of circadian rhythm more generally as this is what can be found in the literature. Coming to the objectives, we have now clarified that chronotype as a proxy for individual circadian rhythm will be measured. This was also changed in the abstract. 

2. Objectives and hypothesis: They indicate three main hypotheses, in the first of these (lines 157-159), they indicate “a depressed patient subgroup. They mean that they will evaluate the subtypes of depression? In any case, to which subgroup do they refer?

I do not find in the introduction the support of this hypothesis.

We reformulated the term “depressed patient subgroup” to make it clearer that we hypothesize to find a pattern of the mentioned variables expressed a certain way, thereby characterizing some of the investigated patients as a new subgroup. Further, an additional sentence was inserted in the objectives to better explain the line of reasoning for analyzing the mentioned variables conjointly. As for reference of the hypothesized variable expressions in association to other variables, please see the introduction. 

3. Materials and Methods: Eligibility criteria. It would be useful if they indicated what are the criteria for hospitalization "psychiatric inpatients". Not all readers are familiar with psychiatric treatments, define acronyms for ECT.

This is a naturalistic observational design. Thus, patients are included in the study as they naturalistically are hospitalized. To get a better impression of this patient group, we have added some very common reasons for why depressed patients may be hospitalized. Abbreviation is now written in full and the whole text was checked again for unexplained acronyms. 

4. Outcomes: The authors only indicate two outcomes: “body weight” and “depression symptom change”, but it is not clear what response they will evaluate with respect to the first hypothesis (lines 157-159) when referring to “a depressed patient subgroup”.

The outcomes for the first hypothesis are all the named variables since analyses will be of correlative nature. We have added an explanation to the outcomes and better clarified primary and secondary outcomes.

---

## [Decision Letter · Decision Letter 1]

8 Jul 2022

A comprehensive approach to predicting weight gain and therapy response in psychopharmacologically treated major depressed patients: a cohort study protocol.

PONE-D-21-24585R1

Dear Dr. Simon,

We’re pleased to inform you that your manuscript has been judged scientifically suitable for publication and will be formally accepted for publication once it meets all outstanding technical requirements.

Kind regards,

Carla Pegoraro

Division Editor

PLOS ONE

Additional Editor Comments (optional):

Reviewers' comments:

Reviewer's Responses to Questions

**Comments to the Author**

1. Does the manuscript provide a valid rationale for the proposed study, with clearly identified and justified research questions?

Reviewer #1: Yes

2. Is the protocol technically sound and planned in a manner that will lead to a meaningful outcome and allow testing the stated hypotheses?

Reviewer #1: Yes

3. Is the methodology feasible and described in sufficient detail to allow the work to be replicable?

Reviewer #1: Yes

4. Have the authors described where all data underlying the findings will be made available when the study is complete?

Reviewer #1: Yes

5. Is the manuscript presented in an intelligible fashion and written in standard English?

Reviewer #1: Yes

6. Review Comments to the Author

You may also provide optional suggestions and comments to authors that they might find helpful in planning their study.

Reviewer #1: The authors have resolved the observations that I made to them, in an appropriate manner. I consider that the document is accepted.

7. PLOS authors have the option to publish the peer review history of their article (what does this mean?). If published, this will include your full peer review and any attached files.

Reviewer #1: **Yes: **SILVIA ARACELY TAFOYA

---

## [Editor Report · Acceptance letter]

12 Jul 2022

PONE-D-21-24585R1 

A comprehensive approach to predicting weight gain and therapy response in psychopharmacologically treated major depressed patients: a cohort study protocol. 

Dear Dr. Simon:

I'm pleased to inform you that your manuscript has been deemed suitable for publication in PLOS ONE. Congratulations! Your manuscript is now with our production department. 

Kind regards, 

on behalf of

Dr Carla Pegoraro 

Staff Editor

PLOS ONE